# Generation of full-length circular RNA libraries for Oxford Nanopore long-read sequencing

Steffen Fuchs[ID][1,2,3,4,5,6]*, Loélia Babin[5,6], Elissa Andraos[5,6], Chloé Bessiere[5,6], Semjon Willier[7], Johannes H. Schulte[1,2,3,4], Christine Gaspin[8,9], Fabienne Meggetto[5,6]*

**1** Department of Pediatric Oncology and Hematology, Charité - Universitätsmedizin Berlin, Berlin, Germany, **2** German Cancer Consortium (DKTK), Partner Site Berlin, Berlin, Germany, **3** German Cancer Research Center (DKFZ), Heidelberg, Germany, **4** Berlin Institute of Health at Charité –Universitätsmedizin Berlin, BIH Biomedical Innovation Academy, BIH Charité Clinician Scientist Program, Berlin, Germany, **5** CRCT, Inserm, CNRS, Université Toulouse III-Paul Sabatier, Centre de Recherches en Cancérologie de Toulouse, Université de Toulouse, Toulouse, France, **6** Laboratoire d'Excellence Toulouse Cancer-TOUCAN, Toulouse, France, **7** Department of Pediatric Hematology, Oncology and Stem Cell Transplantation, Dr. von Hauner Children's Hospital, University Hospital, LMU Munich, Munich, Germany, **8** INRAE, BioinfOmics, GenoToul Bioinformatics Facility, Université Fédérale de Toulouse, Castanet-Tolosan, France, **9** INRAE, MIAT, Université Fédérale de Toulouse, Castanet-Tolosan, France

* steffen.fuchs@charite.de, steffen.fuchs@inserm.fr (SF); fabienne.meggetto@inserm.fr (FM)

**Data Availability Statement:** Sequencing data generated in this study are available at the NCBI Gene Expression Omnibus (GSE197872; https://www.ncbi.nlm.nih.gov/geo). All other data are

## Abstract

Circular RNA (circRNA) is a noncoding RNA class with important implications for gene expression regulation, mostly by interaction with other RNA species or RNA-binding proteins. While the commonly applied short-read Illumina RNA-sequencing techniques can be used to detect circRNAs, their full sequence is not revealed. However, the complete sequence information is needed to analyze potential interactions and thus the mechanism of action of circRNAs. Here, we present an improved protocol to enrich and sequence full-length circRNAs by using the Oxford Nanopore long-read sequencing platform. The protocol involves an enrichment of lowly abundant circRNAs by exonuclease treatment and negative selection of linear RNAs. Then, a cDNA library is created and amplified by PCR. This protocol provides enough material for several sequencing runs. The library is used as input for ligation-based sequencing together with native barcoding. Stringent quality control of the libraries is ensured by a combination of Qubit, Fragment Analyzer and qRT-PCR. Multiplexing of up to 4 libraries yields in total more than 1–2 Million reads per library, of which 1–2% are circRNA-specific reads with >99% of them full-length. The protocol works well with human cancer cell lines. We further provide suggestions for the bioinformatic analysis of the created data, as well as the limitations of our approach together with recommendations for troubleshooting and interpretation. Taken together, this protocol enables reliable full-length analysis of circRNAs, a noncoding RNA type involved in a growing number of physiologic and pathologic conditions.

Metadata

**Associated content**. https://dx.doi.org/10.17504/protocols.io.rm7vzy8r4lx1/v2.

available from the corresponding authors on reasonable request. Additionally, the DOI for the protocol mentioned in the lab protocol paper. It is the following: dx.doi.org/10.17504/protocols.io. cbs9snh6 https://protocols.io/view/generation-of-full-length-circrna-libraries-for-ox-cbs9snh6.html.

**Funding:** S.F. is a participant in the BIH-Charité Clinician Scientist Program funded by the Charité – Universitätsmedizin Berlin and the Berlin Institute of Health. S.F. was supported during work by a grant from the Berliner Krebsgesellschaft e.V. (grant no. FUFF201721KK), the Stiftung Tumorforschung Kopf-Hals (Wiesbaden, Germany) and the Deutsche Forschungsgemeinschaft (DFG, German Research Foundation, grant no. 439441203). F.M was supported by grants from Inserm, l'association Eva pour la vie, the Federation Grandir Sans Cancer and La ligue contre le cancer (Equipes Labelisée 2017-2021). E.A. is supported by a grant from Labex TOUCAN/Laboratoire d'excellence Toulouse Cancer. L.B. and C.B. are supported by fellowships from the Fondation de France. The funders had and will not have a role in study design, data collection and analysis, decision to publish, or preparation of the manuscript.

**Competing interests:** The authors have declared that no competing interests exist.

## Introduction

Circular RNAs (circRNAs) are a class of noncoding RNA, which is generated by a form of alternative splicing termed back-splicing. Their ring-like structure and lack of free 5' and 3' ends render them exonuclease resistant and more stable than linear RNAs. This is the reason why they escape detection by the highly used poly(A)-selected mRNA sequencing [1, 2]. circRNAs regulate gene expression by e.g. binding to microRNAs or RNA-binding proteins, or interact directly with the transcriptional machinery [3–5]. The commonly applied Illumina-based short-read sequencing techniques can be readily used (total RNA-seq) to identify circRNAs by their characteristic back-splice junction [6]. However, since circRNAs, which have an average size ranging from 200 to 800 nt [7, 8], share their remaining sequence with their cognate linear RNAs from the same gene, the full-length information cannot be confidently retrieved by these methods. The recently developed long-read RNA-sequencing techniques such as the Oxford-Nanopore sequencing platform bear great potential to obtain this missing information due to their ability to sequence transcripts in full-length.

Sequencing of circRNAs by a direct RNA-sequencing protocol with Oxford Nanopore is an attractive option to analyze circRNAs including their epigenetic modifications without introduction of bias by PCR. However, such an approach requires a linearization of circRNA molecules and has a reduced sensitivity, since the detection of lowly abundant circRNAs is limited and the sequencing coverage is not as high as for Illumina-based methods. Wang *et al.* followed this approach to analyze plant RNA [9], which they fragmented to be able to sequence it. While several circRNAs were detected, the high amount of input RNA remains a critical limitation in particular when analyzing human and especially patient samples that are often degraded.

Recently, a workflow was published to sequence circRNAs by creating a cDNA library that is amplified by PCR, without the need of fragmentation [8]. The approach of Zhang *et al.* uses a ribodepletion followed by an enrichment of circRNAs by exonuclease treatment and a size selection of transcripts longer than 1 kb. The created library is then used for ligation-based sequencing with Oxford Nanopore. With this approach the team obtained between 0.8 and 4 million reads per library, of which 1–6% were circRNA-specific reads that mapped to the back-splice junction. Here, we present a modified version of this workflow that we adapted to retrieve full-length sequencing information also of shorter circRNAs to cover the whole spectrum of circRNAs. The workflow produces an increased library output to sequence several times, if needed, to potentially detect also lowly abundant circRNAs. In more detail, we changed the ribodepletion method from a commercial kit to the published method of Baldwin *et al.* [10], which worked more efficiently in our hands. This ribodepletion method is based on a pool of DNA oligonucleotides that hybridize with ribosomal RNA and a subsequent digest of DNA:RNA hybrids by RNaseH. For further circRNA enrichment a negative selection of poly (A) transcripts was added by using oligo(dT)-conjugated magnetic beads. The final size selection of the amplified circRNA library was adapted to include shorter circRNAs. Finally, we introduced a quality control by qRT-PCR to detect the enrichment of circRNAs and the depletion of unwanted transcripts, such as ribosomal RNA, mitochondrial RNA and small-nucleolar RNAs. Further, we provide recommendations for the Nanopore library creation and sequencing together with recommendations for the bioinformatics analysis.

## Materials and methods

The protocol described in this peer-reviewed article is published on protocols.io, https://dx.doi.org/10.17504/protocols.io.rm7vzy8r4lx1/v2 **(version 2)** and is included for printing as S1 File with this article.

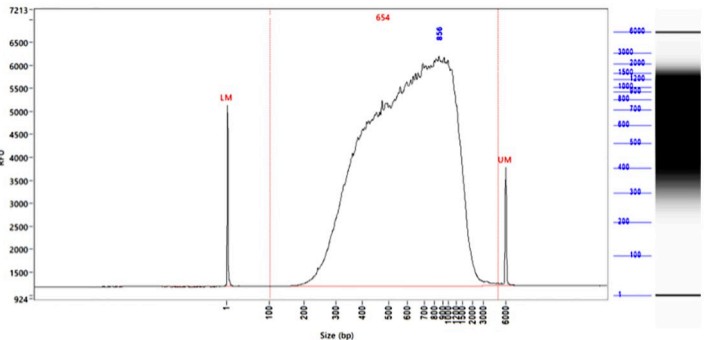

**Fig 1. Generated libraries have the size of the average circRNA length.** Shown is the library created from RNA of the anaplastic large-cell lymphoma cell line SU-DHL-1. The library size was analyzed by Fragment Analyzer with the kit hs NGS. The average library size was 654 nt. RFU, relative fluorescence units.

## circRNA validation

1 µg RNA isolated from COST anaplastic large-cell lymphoma (ALCL) cells was used to create cDNA with the ProtoScript II First Strand cDNA Synthesis kit (New England Biolabs, #E6560S) according to the manufacturer using random hexamer primers. An RT-PCR was performed using the FastStart Taq DNA Polymerase (Roche, #4738381001) according to the manufacturer. The used primer sequences are as follows: circARID1A (F: CTCCAGTAAGG-GAGGGCAAG, R: TGTTGCTGCGAGTATGGGTT), circNFATC3 (F: TGAAACTGAAGGTAGCCGAGG, R: ATGTGGTAAGCAAAGTGGTGTG), circASAP1 (F: GGATACAGCATGCATCAGCTC, R: TCAGCTCCTGTTATCTCTGTGC), circSETD3 (F: TCCTTTGGTGACACAGTTGCT, R: ACTCCTTCTGGCAGCCCTAT) and circZBTB46 (F: CCGGTAGTGGGACGTGATTT, R: ACTCGCTGTCCCAGTCTGTA). PCR products were analyzed by agarose gel electrophoresis (2% agarose, 100 V, 30 min.). The band of the expected size was cut and the DNA cleaned by NucleoSpin Gel and PCR Clean-up kit (Macherey Nagel, #740609.50) following the manufacturer's recommendations. The samples were sent for Sanger sequencing to Eurofins Genomics (Ebersberg, Germany). Obtained sequences were analyzed by NCBI BLAST (https://blast.ncbi.nlm.nih.gov/Blast.cgi) and aligned to the human genome (GRCh38) and the respective circRNA sequence.

## Expected results

Using the described protocol, we prepared sequencing libraries of 4 different anaplastic large-cell lymphoma (ALCL) cell lines that served as a model to test the workflow (SU-DHL-1, Karpas-299, COST [11], SUP-M2). The obtained libraries had an average length of 606.8 nt and a concentration of 4.8 ng/µl (117–133 ng of library in total), respectively (Fig 1, Table 1). The library length corresponds with the average published size of circRNAs, which is between 200–800 nt [7, 8], thus showing that our workflow maintains the size of circRNAs and does not degrade them.

**Table 1. Results of the library preparation.** Libraries for Nanopore sequencing were prepared of 4 anaplastic large-cell lymphoma cell lines (SU-DHL1, Karpas-299, COST, SUP-M2). The concentration was measured with the Qubit BR dsDNA kit and library size by Fragment Analyzer with the hs NGS kit.

| | SU-DHL-1 | Karpas-299 | COST | SUP-M2 | Average |
|---|---|---|---|---|---|
| **Concentration [ng/µl]** | 5.12 | 4.68 | 4.92 | 4.5 | 4.81 |
| **Size [nt]** | 654 | 629 | 573 | 571 | 606.8 |

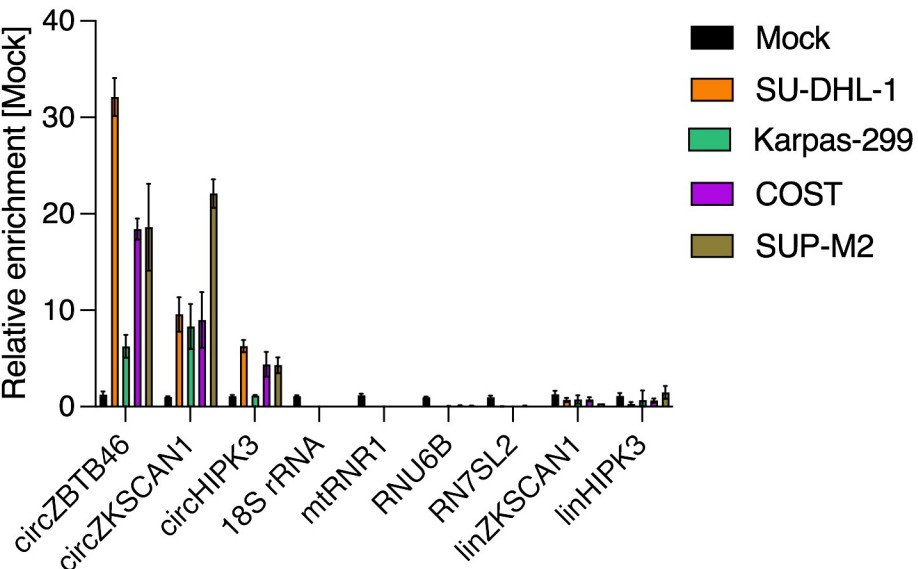

**Fig 2. circRNAs get enriched by the library workflow.** 4 different RNA samples from anaplastic large-cell lymphoma cell lines (SU-DHL-1, Karpas-299, COST and SUP-M2) were treated enzymatically to enrich for circRNAs as described in the protocol. The expression of circRNAs and unwanted transcripts (ribosomal RNA, 18S rRNA; mitochondrial RNA, mtRNR1; small-nucleolar RNA, RNU6B; signal recognition particle RNA, RN7SL2 and linear RNAs/mRNAs, linkZKSCAN1, linHIPK3) was analyzed by qRT-PCR and compared with an untreated Mock control.

As part of our introduced quality control workflow, a qRT-PCR was performed to detect the enrichment of circRNAs and the depletion of unwanted RNA transcripts (Fig 2). We used 3 different circRNAs as indicator for an enrichment, of which we know from previous Illumina-based RNA-sequencing experiments that they are well expressed in our cell line models. The enrichment was on average between 4 and 17-fold in comparison to the non-enriched control. Ribosomal RNA was depleted more than 30,000-fold and also other unwanted transcripts were degraded (mitochondrial RNA, small nucleolar RNA, signal recognition particle RNA, linear RNAs and mRNAs).

The created library pool was used as input for Oxford Nanopore sequencing with the ligation-based sequencing kit (SQK-LSK109) together with the native barcoding kit (EXP-NBD104) according to the manufacturer and sequenced on one flow cell on a MinION MK1C. The sequencing output was on average 1,536,229 reads per library and the reads were of high quality (Table 2, mean Q-score 15).

**Table 2. Sequencing results obtained with one MinION flow cell.** circRNA-enriched libraries from 4 anaplastic large-cell lymphoma cell lines were sequenced by Oxford Nanopore. Calculations are based on the passed reads and the circRNA analysis was performed with CIRI-Long. BSJ, back-splice junction.

|  | SU-DHL1 | Karpas-299 | COST | SUP-M2 | Average |
|---|---|---|---|---|---|
| **Raw reads** | 1,473,419 | 1,734,196 | 899,725 | 2,037,577 | 1,536,229 |
| **Mean read length [nt]** | 459.6 | 368.2 | 386.3 | 403.3 | 404.4 |
| **Maximum read length [nt]** | 4,006 | 3,889 | 3,538 | 3,455 | 3,722 |
| **BSJ-reads (% of reads)** | 1.05 | 0.95 | 0.95 | 1.06 | 1.00 |
| **Full-length circRNAs** | 15,673 | 16,725 | 8,750 | 21,918 | 15,767 |
| **Different circRNAs** | 3,143 | 3,195 | 1,426 | 4,017 | 2,945 |
| **Mean circRNA length [nt]** | 435.1 | 354.9 | 366.4 | 370.4 | 381.7 |
| **Maximum circRNA length [nt]** | 1,798 | 1,634 | 1,596 | 2,228 | 1,814 |

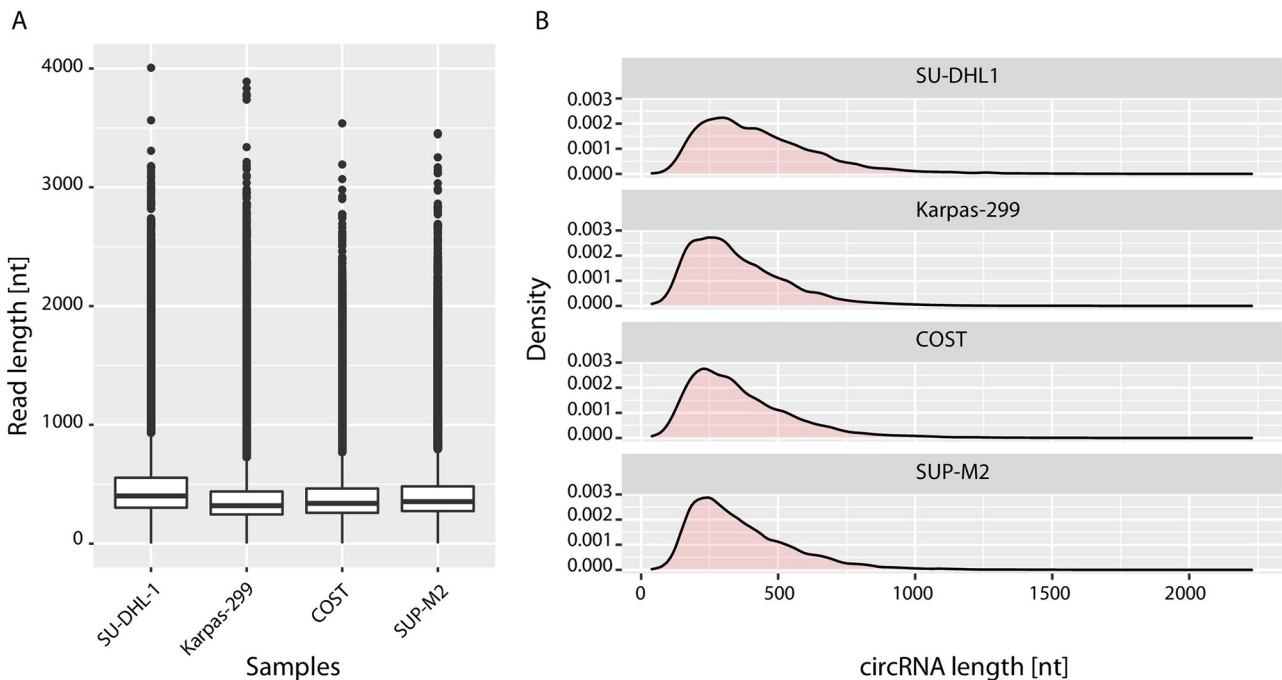

**Fig 3. Distribution of the length of identified circRNAs.** Shown are the results of circRNA sequencing from 4 anaplastic large-cell lymphoma (ALCL) cell lines with Oxford Nanopore. A) Distribution of read length and B) length of the identified circRNAs in the 4 different ALCL cell line samples. Gaussian kernel density estimation was used to calculate the distribution of circRNA length.

Importantly, the pores were not completely saturated, so probably a longer sequencing run with more material would have been possible. Base calling of the raw sequencing data was performed with Guppy from the MinKNOW software (v22.05.8, part of the operation system of the MinION sequencing device) and fastq files were generated. Bioinformatics analysis of the fastq files involved cleaning the reads from adapter sequences with cutadapt (v.3.4, https://doi.org/10.14806/ej.17.1.200). Then the CIRI-Long software (v1.0.3, [8]) was used to detect circRNAs using default settings (detailed recommendations are included in our protocol as S1 File). Following the analysis workflow described for CIRI-Long in our protocol we could identify on average 15,767 circRNA-specific reads, thus 1.0% of the total reads, of which 99% covered the full length of the circRNA (Figs 3 and 4), similar to the study from Zhang *et al.* [8]. On average 2,945 different circRNAs were identified. Of note, it is visible that when more reads are generated, more different full-length circRNA isoforms are detected, which could be another argument for deeper sequencing. The results were comparable among the samples from the 4 different human cancer cell lines, which demonstrates the robustness of the workflow.

We then selected randomly 5 different circRNAs detected by Nanopore for validation (Fig 5 and S1 Fig). COST ALCL cells were used to isolate RNA and transcribe cDNA. Primers were designed to specifically amplify the circRNAs, which were submitted for Sanger sequencing. The characteristic back-splice junction essential for the formation of the RNA circle was detected for all of the circRNAs, which validates our sequencing workflow.

The limitations of this protocol involve the relatively low abundance of circRNAs in general. Since Nanopore sequencing creates less reads in comparison to Illumina-based techniques, especially lowly abundant circRNAs might not be detected by our protocol. Further, the recommended RNA input of 7 µg to have enough material for several rounds of

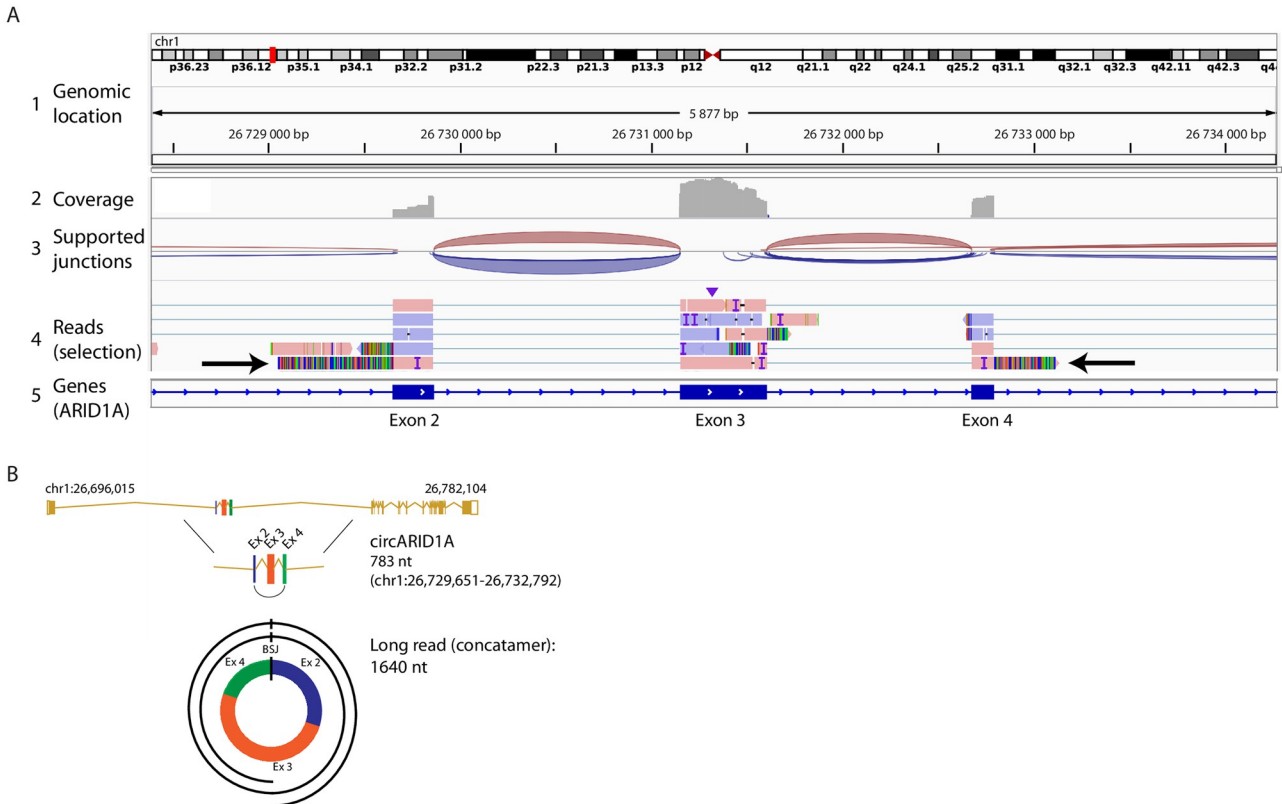

**Fig 4. One long read captures the entire length of one circRNA several times.** A) Shown is exemplified the alignment of sequencing reads obtained by Oxford Nanopore sequencing in the anaplastic large-cell lymphoma cell line SU-DHL-1. The alignment to exon 2–4 of the *ARID1A* gene supports a circRNA with a back-splice junction between exon 4 and 2 that was captured by one long-read (arrows). Track 1: genomic location, track 2: sequencing coverage, track 3: predicted splice junctions, track 4: selected sequencing reads and their alignment, track 5: genes and their exons and introns. The alignment was visualized with IGV Genomics Viewer. B) Scheme showing the long read from panel A aligning several times to circARID1A derived from *ARID1A*. Marked is the back-splice junction (BSJ).

sequencing and also detect lower abundant circRNAs might be too high for situations, where the material is limited. However, we successfully tested our protocol with as little as 3 µg, without impairing the sequencing output. If the obtained amount of DNA by the circRNA enrichment workflow is not high enough, then the PCR amplification of the cDNA libraries can be adapted by increasing the volume of the PCR reaction and the number of cycles (step 12 of the protocol). Further, if the expected size of circRNAs in the cell model of interest is lower or higher than 200–800 nt, the size selection (step 13 of the protocol) can be easily adapted by changing the ratio of beads to DNA (lower ratio selects for larger fragments, higher ratio selects for shorter fragments). Limitations concerning the used tool CIRI-Long to detect circRNAs are that the tool does not detect circRNAs derived from fusion genes. Further, the alignment parameters are not modifiable and a bam file containing the aligned reads is not conserved. Therefore, we routinely perform a separate alignment using minimap2 (v.2.19, [12]) and a visualization by IGV Genomics Viewer (v2.9.4, [13]). By this means, chimeric alignments containing segments of the same read aligning to distant genes can be visualized and help to identify fusion gene derived circRNAs.

In summary, this protocol facilitates consistent full-length sequencing of circRNAs, which will help to study this noncoding RNA type in a variety of physiologic and pathologic contexts.

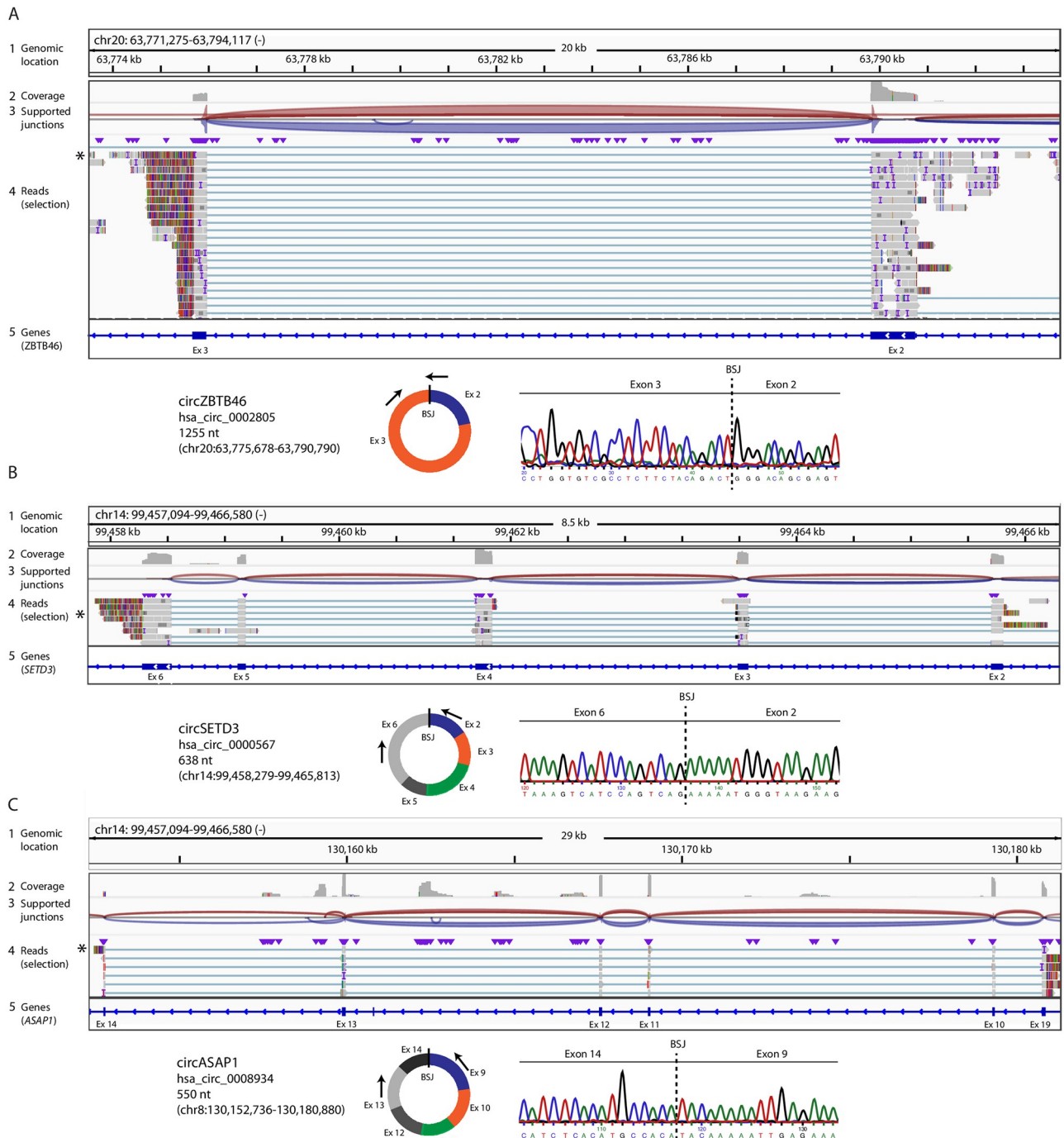

**Fig 5. circRNAs detected by Nanopore were validated by Sanger sequencing.** 3 different circRNAs (A: circZBTB46, B: circSETD3, C: circASAP1) detected by Nanopore-seq were amplified by RT-PCR using cDNA from COST ALCL cells and analyzed by Sanger sequencing. For each panel, the alignment of long reads against the exons that form the circRNA is shown in the upper part as visualized by IGV Genomics Viewer (tracks as in Fig 3). In the lower part a scheme of the circRNA is shown, together with the localization of the used primers and the BSJ-sequence obtained by Sanger sequencing. The circbase.org ID is mentioned [14]. BSJ, back-splice junction.

## Supporting information

**S1 Fig. Validation of circRNAs by Sanger sequencing.** 2 circRNAs detected by Nanopore-seq were validated by Sanger sequencing similar as in Fig 5. Representative alignments and the BSJ-sequence obtained by Sanger sequencing are shown for A) circNFATC3 and B) circAR-ID1A (alignments see Fig 4). The circbase.org ID is mentioned [14]. BSJ, back-splice junction.
(TIF)

**S1 File. Step-by-step protocol, also available on protocols.io.**
(PDF)

## Acknowledgments

The authors are grateful to Emeline Sarot and Nathalie Saint-Laurent (Genomic and Transcriptomic facility, Technology Cluster of the Cancer Research Center of Toulouse, INSERM-UMR1037) for their technical assistance. The authors thank further Falk Hertwig and Filippos Klironomos (Charité, Berlin) for help during the establishment of this protocol.

## Author Contributions

**Conceptualization:** Steffen Fuchs, Fabienne Meggetto.

**Data curation:** Steffen Fuchs, Loélia Babin, Christine Gaspin.

**Formal analysis:** Steffen Fuchs, Loélia Babin, Christine Gaspin.

**Funding acquisition:** Steffen Fuchs, Fabienne Meggetto.

**Investigation:** Steffen Fuchs, Loélia Babin, Elissa Andraos, Chloé Bessiere, Semjon Willier, Johannes H. Schulte, Christine Gaspin.

**Methodology:** Steffen Fuchs.

**Project administration:** Steffen Fuchs, Fabienne Meggetto.

**Resources:** Fabienne Meggetto.

**Software:** Loélia Babin, Chloé Bessiere, Christine Gaspin.

**Supervision:** Steffen Fuchs, Fabienne Meggetto.

**Validation:** Steffen Fuchs, Loélia Babin, Christine Gaspin.

**Visualization:** Steffen Fuchs, Loélia Babin, Christine Gaspin.

**Writing – original draft:** Steffen Fuchs, Fabienne Meggetto.

**Writing – review & editing:** Steffen Fuchs, Fabienne Meggetto.

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
