## [Decision Letter · Decision Letter 0]

23 May 2022

PONE-D-22-06858Generation of full-length circular RNA libraries for Oxford Nanopore long-read sequencingPLOS ONE

Dear Dr. Fuchs,

Thank you for submitting your manuscript to PLOS ONE. After careful consideration, we feel that it has merit but does not fully meet PLOS ONE’s publication criteria as it currently stands. Therefore, we invite you to submit a revised version of the manuscript that addresses the points raised during the review process. Among all comments, please take into account those regarding the methodology, challenges and interpretations.In that sense, the whole protocol must be explained in detail.

We look forward to receiving your revised manuscript.

Kind regards,

Eduardo Andrés-León

Academic Editor

PLOS ONE

Journal Requirements:

2. Thank you for providing the following Protocols.io DOI in your submission form [Protocols.io DOI]. In keeping with our submission requirements, please add the Protocols.io DOI to the Methods section of your manuscript as well using this format: “The protocol described in this peer-reviewed article is published on protocols.io, https://dx.doi.org/10.17504/protocols.io[........] and is included for printing as supporting information file 1 with this article.” For more information, please see our submission guidelines: https://journals.plos.org/plosone/s/submission-guidelines#loc-guidelines-for-specific-study-types

Reviewers' comments:

Reviewer's Responses to Questions

**Comments to the Author**

1. Does the manuscript report a protocol which is of utility to the research community and adds value to the published literature?

Reviewer #1: Yes

Reviewer #2: Yes

2. Has the protocol been described in sufficient detail?

Descriptions of methods and reagents contained in the step-by-step protocol should be reported in sufficient detail for another researcher to reproduce all experiments and analyses. The protocol should describe the appropriate controls, sample sizes and replication needed to ensure that the data are robust and reproducible.

Reviewer #1: No

Reviewer #2: Yes

3. Does the protocol describe a validated method?

Reviewer #1: No

Reviewer #2: Yes

4. If the manuscript contains new data, have the authors made this data fully available?

Reviewer #1: No

Reviewer #2: Yes

**5. Is the article presented in an intelligible fashion and written in standard English?**

Reviewer #1: Yes

Reviewer #2: Yes

6. Review Comments to the Author

Reviewer #1: In this manuscript, the authors have developed a circRNA sequencing protocol that allows full length analysis using Oxford nanopore. Overall, the attempt to identify the full length of circRNAs is appreciated. However, the authors need to include the full detailed protocol within this manuscript with full analysis, troubleshooting, challenges, expected results and interpretations. The authors should also provide other data than those published at protocols.io with examples of fully delineated sequences of various circRNAs and confirmed by sanger sequencing.

Reviewer #2: Fuchs et al. present a comprehensive protocol based on the published Full-length sequencing protocol by Zhang et al. The adapted approach differs in several aspects, such as the optimization of different circRNA lengths and addition of cleanup procedures as well as QC steps.

1. The authors mention “Modification of the ribodepletion method” as one of the parts changed in the adapted protocol. A short description on what the change is would be helpful in the abstract part of the protocol.

2. The protocol suggests use of qRT-PCR for validation of circRNAs and linear RNAs. It might be helpful to include a link to tools that can easily generate circRNA-specific primer pairs, as standard tools usually do not easily work with circular RNAs.

3. It would be helpful to directly state in the abstract on protocols.io the required amount of starting material, as this is crucial in cases where only little material is available.

4. In general, the generated data data is freely available, however, the GEO entry has an embargo date of March 2023, thus the data is not immediately available.

7. PLOS authors have the option to publish the peer review history of their article (what does this mean?). If published, this will include your full peer review and any attached files.

Reviewer #1: No

Reviewer #2: No

---

## [Author Response · Author response to Decision Letter 0]

25 Jun 2022

Dear Dr. Andrés-León,

Thank you very much for the consideration of our manuscript and the assessment. Further, we would like to thank you for the deadline extension, which allowed us to perform a thorough review. We appreciate the comments of the two reviewers that were very helpful for us. We carefully took their concerns and recommendations into account and now provide a much more detailed protocol (version 2), including further information about the bioinformatics analysis, challenges and limitations together with expected results of this protocol and their interpretation. Further, we created new in vitro data to show the robustness of our approach by validating several circRNAs detected by Nanopore via Sanger sequencing. Changes in the manuscript are marked in blue and are italicized. We updated our protocol on protocols.io and attached a .pdf copy as supplementary file to the manuscript. The changes in the protocol we made are described in detail below, since the protocols.io website does not allow tracking of the changes.

We hope that the following explanations as well as our point-by-point response to the reviewers, where we have addressed all of their concerns and comments, will make the manuscript acceptable for publication in PLOS ONE. 

If you require any further information, please do not hesitate to contact us directly. Thank you for your time and consideration. We look forward to hearing from you at your earliest convenience. 

Yours sincerely, 

Steffen Fuchs

 

Reviewer #1: 

In this manuscript, the authors have developed a circRNA sequencing protocol that allows full length analysis using Oxford nanopore. Overall, the attempt to identify the full length of circRNAs is appreciated. However, the authors need to include the full detailed protocol within this manuscript with full analysis, troubleshooting, challenges, expected results and interpretations. The authors should also provide other data than those published at protocols.io with examples of fully delineated sequences of various circRNAs and confirmed by sanger sequencing.

Response:

We thank the reviewer for the appreciation of our provided approach to sequence circRNAs in full length and for the provided feedback and comments. We agree with the reviewer that more methodical details of the protocol, including its limitations and challenges together with troubleshooting and further more details on the expected results and their interpretation will be very helpful for the researchers planning to use it. We therefore expanded our protocol and added those details to “Steps” and added these new sections to the protocol: 5) Suggestions for Nanopore sequencing, 6) Recommendations for bioinformatics analysis of the data, 7) Expected results and interpretation, 8) Limitations and challenges, 9) Troubleshooting. 

Although, the creation of the sequencing libraries and the sequencing itself follows the official protocol provided by Oxford Nanopore (kits EXP-NBD104 and SQK-LSK109), we provide now explanations and suggestions for modifications that improved sequencing for us. For instance, we included an optional part on washing the flow cell during a sequencing run and reloading the library to obtain more sequencing reads. 

While the focus of this protocol lies on the generation of the enriched circRNA-fraction for library generation, we agree with the reviewer that more information concerning the data analysis should be provided. Therefore, we provide now detailed recommendations concerning the bioinformatics analysis including the used commands, the limitations of the analysis and how to overcome them (in the protocol at section 6) “Recommendations for bioinformatics analysis of the data” and in sections 7 to 9).

This protocol is part of the materials and methods section of the manuscript and further attached as supplementary data file, now in the more detailed version 2. In the manuscript we added a summary of all this information in the abstract lines 71-73 (page 3), the introduction at lines 117-118 (page 5), and in the expected results section at lines 189-194 (page 8) and lines 236-254 (pages 9, 10).

We appreciate the recommendation of the reviewer to validate circRNAs by Sanger sequencing that we detected by Oxford Nanopore. This approach will help to show the robustness of our workflow. We now selected randomly 5 circRNAs that were detected by Nanopore. The circRNAs were amplified by PCR and Sanger sequencing was performed. With this approach we could confirm all of the selected circRNAs as indicated by the detected back-splice junction in one of the anaplastic large-cell lymphoma cell lines that we sequenced by Nanopore. This information is now added as Figure 5 and supplementary figure S1 in the manuscript. Text was added to the manuscript in lines 124-140 (pages 5, 6), 217-234 (pages 8, 9).

Reviewer #2: 

Fuchs et al. present a comprehensive protocol based on the published Full-length sequencing protocol by Zhang et al. The adapted approach differs in several aspects, such as the optimization of different circRNA lengths and addition of cleanup procedures as well as QC steps.

1. The authors mention “Modification of the ribodepletion method” as one of the parts changed in the adapted protocol. A short description on what the change is would be helpful in the abstract part of the protocol.

Response:

We thank the reviewer for this comment. Indeed, the ribodepletion method is crucial for the enrichment of circRNAs for the generation of libraries, since their abundance is much lower than that of ribosomal RNA. In the basic protocol from Zhang et al. the authors propose the use of a commercial ribodepletion kit (RiboErase kit, #07962266001, Kapa Biosystems). This kit, like other newer kits e.g. from New England Biolabs (NEBNext rRNA Depletion Kit v2, #E7405), is based on the protocol of Adiconis et al.[1], which uses a pool of DNA oligonucleotides directed against human rRNAs followed by a digest of DNA:RNA hybrids by RNaseH. However, the exact composition of the commercial kits remains proprietary and the used sequences are not public. In our protocol, we use the method published by Baldwin et al. [2] that is an updated and more efficient version of the Adiconis method. We use this method routinely as well successfully for the creation of Illumina RNA-sequencing libraries. Key changes are an increased ratio of DNA oligonucleotides to RNA (5:1, whereas in the Adiconis protocol a 1:1 ratio is used) and a higher incubation temperature of RNaseH (65°C in comparison to 45°C), which increases the activity and reduces the incubation time. We compared the commercial kit with the Adiconis and the Baldwin method. Adiconis and Baldwin’s methods outperformed the kit, while Baldwin’s method led to the most efficient ribodepletion. We added this information to the manuscript in line 110-112 (page 5) and in the protocol in the section “Steps”.

2. The protocol suggests use of qRT-PCR for validation of circRNAs and linear RNAs. It might be helpful to include a link to tools that can easily generate circRNA-specific primer pairs, as standard tools usually do not easily work with circular RNAs.

Response:

We thank the reviewer for this helpful remark. Indeed, it is important to carefully design divergent primers to specifically amplify the backsplice-junction of a circRNA without amplifying the cognate linear RNA transcribed from the same gene. We added information in the protocol to section “4) Quality control, Step 15 Validation of circRNA enrichment” on how to design primers to specifically detect circRNAs and linked two tools for primer design: CircInteractome [3] and CircPrimer 2.0 [4].

3. It would be helpful to directly state in the abstract on protocols.io the required amount of starting material, as this is crucial in cases where only little material is available.

Response:

We appreciate this useful remark of the reviewer and we agree that it is crucial to mention the amount of needed starting material to prepare the sequencing libraries, especially when it comes to patient samples, which might be limited. We tested different RNA quantities to enrich for circRNAs and found that 7 µg, as we wrote in the protocol, works best and further provides enough material to sequence the samples several times to create enough data. However, we also tried 3-5 µg as input, and we could observe no major change in the amount of obtained reads, especially when multiplexing several samples, which should still lead to sufficient pore occupancy while sequencing. We added this information to the abstract and the materials section of the protocol and further in the “Expected results” part of the manuscript, lines 238-242 (page 9).

4. In general, the generated data are freely available, however, the GEO entry has an embargo date of March 2023, thus the data is not immediately available.

Response:

We thank the reviewer for this comment. The data was uploaded to the public repository NCBI GEO and is freely available, thus following the recommendations of PLOS ONE. However, we put an embargo on the dataset for the moment, to protect our data while the manuscript is still under revision. The data will be open as soon as the manuscript is accepted, which is in line with the NCBI GEO recommendations. We are happy to provide a temporary reviewer access to the data for the purpose of the review, which is as follows: NCBI GEO: https://www.ncbi.nlm.nih.gov/geo/, dataset ID: GSE197872, temporary reviewer password: erkpwyakjfezzed.

New references added: 

1. Adiconis X, Borges-Rivera D, Satija R, DeLuca DS, Busby MA, Berlin AM, et al. Comparative analysis of RNA sequencing methods for degraded or low-input samples. Nat Methods. 2013;10(7):623-9. doi: 10.1038/nmeth.2483. PubMed PMID: 23685885; PubMed Central PMCID: PMCPMC3821180.

2. Baldwin A, Morris AR, Mukherjee N. An Easy, Cost-Effective, and Scalable Method to Deplete Human Ribosomal RNA for RNA-seq. Curr Protoc. 2021;1(6):e176. Epub 2021/06/25. doi: 10.1002/cpz1.176. PubMed PMID: 34165268.

3. Dudekula DB, Panda AC, Grammatikakis I, De S, Abdelmohsen K, Gorospe M. CircInteractome: A web tool for exploring circular RNAs and their interacting proteins and microRNAs. RNA biology. 2016;13(1):34-42. doi: 10.1080/15476286.2015.1128065. PubMed PMID: 26669964; PubMed Central PMCID: PMCPMC4829301.

4. Zhong S, Wang J, Zhang Q, Xu H, Feng J. CircPrimer: a software for annotating circRNAs and determining the specificity of circRNA primers. BMC bioinformatics. 2018;19(1):292. doi: 10.1186/s12859-018-2304-1. PubMed PMID: 30075703.

---

## [Decision Letter · Decision Letter 1]

5 Aug 2022

Generation of full-length circular RNA libraries for Oxford Nanopore long-read sequencing

PONE-D-22-06858R1

Dear Dr. Fuchs,

We’re pleased to inform you that your manuscript has been judged scientifically suitable for publication and will be formally accepted for publication once it meets all outstanding technical requirements.

Kind regards,

Eduardo Andrés-León

Academic Editor

PLOS ONE

Additional Editor Comments (optional):

Reviewers' comments:

Reviewer's Responses to Questions

**Comments to the Author**

1. Does the manuscript report a protocol which is of utility to the research community and adds value to the published literature?

Reviewer #1: Yes

Reviewer #2: Yes

2. Has the protocol been described in sufficient detail?

Descriptions of methods and reagents contained in the step-by-step protocol should be reported in sufficient detail for another researcher to reproduce all experiments and analyses. The protocol should describe the appropriate controls, sample sizes and replication needed to ensure that the data are robust and reproducible.

Reviewer #1: Yes

Reviewer #2: Yes

3. Does the protocol describe a validated method?

Reviewer #1: Yes

Reviewer #2: Yes

4. If the manuscript contains new data, have the authors made this data fully available?

Reviewer #1: Yes

Reviewer #2: Yes

**5. Is the article presented in an intelligible fashion and written in standard English?**

Reviewer #1: Yes

Reviewer #2: Yes

6. Review Comments to the Author

Reviewer #1: The authors have adressed my comments. They have expanded on the protocol, added more information regarding the method and also bioinformatics. They also have performed RT-qPCR validation. Please accept with or without changes.

Reviewer #2: The authors addressed all of my points, I have no further questions. Congratulations on a very helpful protocol.

7. PLOS authors have the option to publish the peer review history of their article (what does this mean?). If published, this will include your full peer review and any attached files.

Reviewer #1: **Yes: **Kotb Abdelmohsen

Reviewer #2: No

---

## [Editor Report · Acceptance letter]

26 Aug 2022

PONE-D-22-06858R1 

Generation of full-length circular RNA libraries for Oxford Nanopore long-read sequencing 

Dear Dr. Fuchs:

I'm pleased to inform you that your manuscript has been deemed suitable for publication in PLOS ONE. Congratulations! Your manuscript is now with our production department. 

Kind regards, 

on behalf of

Dr. Eduardo Andrés-León 

Academic Editor

PLOS ONE